# Application of Metal Nanoparticles for Production of Self-Sterilizing Coatings

**Dariusz Góral** [1],[*] and **Małgorzata Góral-Kowalczyk** [2]

1 Department of Biological Bases of Food and Feed Technologies, Faculty of Production Engineering, University of Life Sciences in Lublin, 28 Głęboka Str., 20-612 Lublin, Poland

2 Department of Agricultural Forestry and Transport Machines, Faculty of Production Engineering, University of Life Sciences in Lublin, 28 Głęboka Str., 20-612 Lublin, Poland; malgorzata.goral-kowalczyk@up.lublin.pl

* Correspondence: dariusz.goral@up.lublin.pl

**Abstract:** Metal nanoparticles (NPs) are increasingly being used in many areas, e.g., industry, pharmacy, and biomedical engineering. NPs can be obtained through chemical and biological synthesis or using physical methods. AgNPs, AuNPs, CuNPs, FeNPs, MgNPs, $SnO_2$NPs, $TiO_2$NPs, and ZnONPs are the most commonly synthesized metal nanoparticles. Many of them have anti-microbial properties and documented activity supported by many tests against some species of pathogenic bacteria, viruses, and fungi. AgNPs, which are used for the production of commercial self-sterilizing packages, are one of the best-explored nanoparticles. Moreover, the EFSA has approved the use of small doses of silver nanoparticles (0.05 mg Ag·kg$^{-1}$) to food products. Recent studies have shown that metal NPs can be used for the production of coatings to prevent the spread of the SARS-CoV-2 virus, which has caused the global pandemic. Some nanoparticles (e.g., ZnONPs and MgONPs) have the Generally Recognized As Safe (GRAS) status, i.e., they are considered safe for consumption and can be used for the production of edible coatings, protecting food against spoilage. Promising results have been obtained in research on the use of more than one type of nanometals, which prevents the development of pathogen resistance through various mechanisms of inactivation thereof.

**Keywords:** metal nanoparticles; coatings; anti-microbial; self-sterilizing; packages

## 1. Introduction

Microbiological stability is particularly important in the case of food with a high water content (e.g., fruit and vegetables). Various physical and chemical methods are used to ensure the microbiological purity of food. There is a trend towards minimization of the use of chemical preservatives and replacement thereof with natural substances. However, this does not always yield a fully satisfactory solution. The great interest in nanoparticles can be attributed to their physicochemical properties and wide application in many industries [1–5]. In agricultural and production practice, cheap and easy protection measures against microorganisms are still a big challenge. New protection strategies against harmful microorganisms are continuously being developed. A promising approach may be the use of nanoparticles with antimicrobial and antiviral properties for the production of face masks, textiles, and other coatings [6–8]. The use of metal nanoparticles for surface functionalization has been particularly successful in obtaining unique textiles with desirable properties, such as self-cleaning, antimicrobial, antifungal, flame retardant, ultraviolet blocking, and superhydrophobicity [9]. The development of novel coatings that are safe, intelligent, and active is an important factor contributing to a reduction in food spoilage. The task of food coatings is to maintain the quality and properties of food products for as long as possible. Various additives are used to obtain intelligent and active food coatings. Nanoparticles occupy a special place among the huge variety of additives [10,11]. The term "nanoparticles" refers to solid particles with a size between 1 and 100 nm [12,13]. Nanoparticles usually have different properties than their macroscopic counterparts. In

biological applications, the reduction in the size of metal particles to the nanometer scale is associated with an increase in their cytotoxicity. This is associated with their larger active surface and thus a higher reactivity than that of conventional materials and interactions with other compounds present in the environment. They also exhibit a higher bioavailability, which makes them easier to adsorb in specific organs, tissues, and cells [14]. Another special active property of nanoparticles is their ability to absorb ultraviolet light, which is considered important for the prevention of photochemical reactions leading to the spoilage of food products. Nanoparticles also exert an antioxidant effect, which is a highly desirable property [10]. Moreover, their most noticeable trait is the lower light scattering degree accompanying a reduction in the particle size. Sufficiently small particles can form transparent coatings. Nanoparticles do not directly induce transparency, but their light scattering effect declines with the decreasing particle size. At the same time, light scattering depends on the difference in the refractive index between the particle and the surrounding medium. A close match of refractive indices promotes the formation of transparent mixtures [12]. Particles must be very well dispersed to achieve transparency in the nanoscale system, as agglomerated nanoparticles have the same optical properties as particles with the size of the agglomerate [12].

Depending on their shape, nanoparticles are classified as zero-dimensional (0D), one-dimensional (1D), two-dimensional (2D), or three-dimensional (3D). This classification is associated with the movement of electrons in different planes. Electrons in 0D are trapped in a dimensionless space, whereas the electrons of 1D nanomaterials can move along the x-axis. 2D and 3D nanomaterials have electrons moving along the x-y axis and the x-y-z axis, respectively [15]. One-dimensional nanomaterials form long nanostructures with thick membranes, i.e., nanotubes, nanofibers, nanowires, nanorods, and nanofilaments. Their length is greater than their width [16]. In turn, 2D nanomaterials have sheet-like structures. These nanomaterials have the largest specific surface area of all known nanoparticles [17]. Due to their high surface-to-volume ratio and the anisotropic physical and chemical properties, 2D nanomaterials are widely used against microorganisms and are attractive for food packaging applications. Large-area 2D nanomaterials directly interact with bacterial membranes, thereby enhancing the antibacterial effect [18,19].

## 2. Aim of the Application of Nanoparticles in Packaging

Nanotechnological inventions are used in computers, nanodiodes, nanotransistors, etc. This technology provides innovative solutions for the production of batteries, fuel cells, and solar cells, which are lighter, more durable, and stronger than those produced with other methods [20]. Nanomaterials exhibit increased catalytic activity, thermal conductivity, and chemical stability [21]. They are also used in various fields of science; in particular, biology and biomedicine. The size of nanorods, nanofibers, and other nanomaterials is comparable to the size of biological particles [22]. Metal-based nanoparticles effectively inactivate microorganisms. Hence, some of them, e.g., silver, titanium dioxide, and zinc oxide nanoparticles, are used as disinfectants and additives in food, cosmetic, and pharmaceutical packaging [14]. Another example is the use of nanoparticles for surface protection and the conservation of antiquities made of marble, stone, glass, bronze, and even objects made of bread [23].

One of the very promising approaches to fighting bacteria is the use of nanomaterial-based antimicrobials. Antimicrobial coatings can be divided into two main groups, germicidal (active) and bacteriostatic (passive). Bactericidal active coatings are able to disrupt the structural integrity of the bacterial cell or membrane. In contrast, bacteriostatic coatings can prevent or reduce the attachment of microorganisms to their surface [24]. Nanoparticles have been well documented to kill bacteria via several different mechanisms [25]. They have the ability to damage the cell membrane physically. They can also generate reactive oxygen species (ROS) and free radicals. This process intensifies oxidative stress, which enhances the fragmentation of genomic DNA and damages the structural integrity of cells [26]. Furthermore, nanoparticles penetrate cell membranes much more easily than

antibiotics [27,28]. Recent reports also indicate that nanoparticle-based antimicrobials act as efficient inhibitors of efflux pumps (proteins present in cell membranes) [29,30]. This proves that nanometals may have multiple applications as antimicrobial agents. Moreover, they are more effective than other antimicrobial agents and are less prone to induce bacterial resistance than antibiotics available on the market.

### 3. Methods for Production of Metal Nanoparticles

Metal nanoparticles can be produced using chemical, physical, and biological methods (Figure 1).

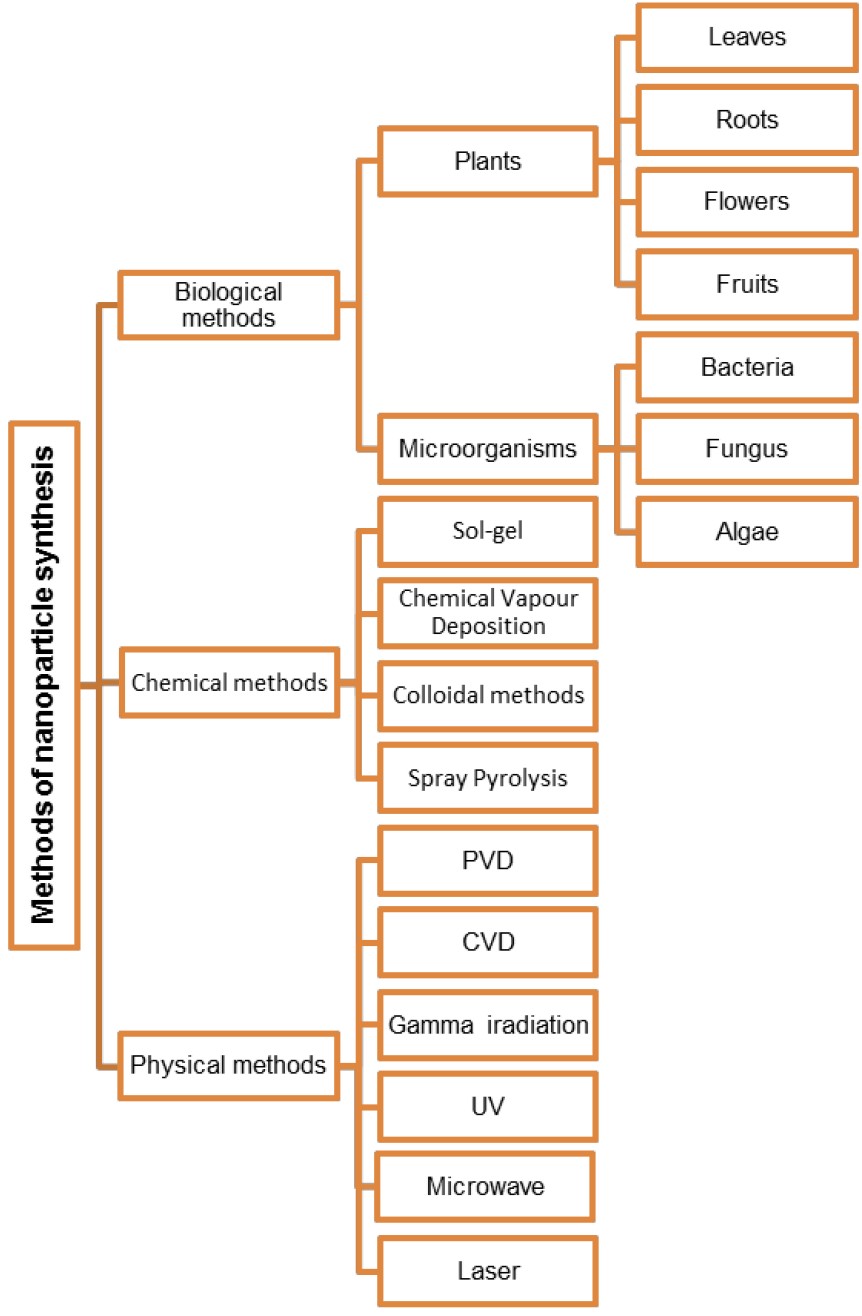

**Figure 1.** Methods for production of metal nanoparticles.

There are two basic strategies used in the production of nanoscale structures: bottom-up and top-down. The former approach is based on the atom-by-atom construction of a nanoparticle. Depending on the required properties of the end product, atoms, molecules,

or colloidal particles can be used as substrates. The control of the synthesis conditions and the initial size of the building material can determine the traits of the final structure. A reverse mechanism is employed in the top-down process based on dispersion methods, which are based on the fragmentation of materials, yielding a final size of the particles in the range of 1–100 nm [31].

### 3.1. Chemical and Physicochemical Methods

Chemical methods are currently regarded as the most popular approaches, although they are not always safe and environmentally friendly. The synthesis of metal nanoparticles using chemical methods is based primarily on chemical reduction. The reaction consists of three steps: reduction of metal salts and formation of free atoms, nucleation, i.e., formation of stable 1–2 nm nuclei, and addition of stabilizing agents to complete the reaction. Currently, this is the cheapest method. Borohydride, citrate, ascorbate, and compounds with hydroxyl or carboxyl groups, i.e., alcohols, aldehydes, carbohydrates, and their derivatives, are the most frequently used reducing agents [32–36]. The process of chemical reduction requires the use of a stabilizing substance whose main task is to protect newly formed nanoparticles against aggregation into larger aggregates [31,37,38]. Such polymers as poly(vinylpyrrolidone) (PVP) [39], poly(methylacrylic acid) (PMAA) [40], poly(methyl methacrylate) (PMMA) [41], and poly(ethylene glycol) (PEG) are used most frequently as stabilizing agents [42]. The greatest advantage of this method is the possible impact on the properties of the final product through regulation of the process conditions. For example, the shape, size, aggregation state, and stability of nanoparticles can be controlled by changing the concentration of the stabilizer or salt, the molar ratio of the reducing and stabilizing agent, temperature, and medium pH [43].

### 3.2. Physical Methods

Physical methods are based on the top-down strategy. Particularly popular are the methods of physical vapor deposition (PVD), in which, a solid material is converted into gas in physical processes and then deposited on the substrate [44]; chemical vapor deposition (CVD), in which, products of chemical reactions are deposited on the substrate [45]; the use of the electromagnetic microwave field, in which, a substantially faster heating rate of the reaction mixture is achieved than in the convection heating process [46]; and the method based on $\gamma$ or UV radiation used as a catalyst for a chemical reaction [47,48]. Another method for the production of various types of nanoparticles is the application of laser radiation. In this approach, nanoparticles are generated by the nucleation and growth of laser-vaporized metals in background gas [49] (Figure 1).

### 3.3. Green Synthesis

Green synthesis is gaining increasing support as a sustainable, reliable, and environmentally friendly approach to the synthesis of various nanoparticles. It does not require the use of solvents that pose a threat to humans and the environment or additional reducing agents. This technique of producing nanoparticles is simple and cost-effective [50–53]. It is based on the production of nanoparticles with the use of plant parts or microorganisms (Figure 1, Table 1) [54]. As in the chemical and physical methods, two approaches in the generation of nanoparticles from living organisms are used, i.e., the top-down and bottom-up strategies. In the top-down methods, the material is fragmented with the use of various techniques, whereas the bottom-up methods are based on the self-organization of atoms leading to the formation of new cell nuclei in chemical or biological reactions [55]. Nanoparticles produced via green synthesis are less toxic to living organisms [56].

**Table 1.** Shape and size of nanoparticles obtained by green synthesis.

| Nanoparticles | Shape | Size, nm | Source | References |
|---|---|---|---|---|
| ZnONPs | spherical | 129 | Daphne leaves | [57] |
| | plate | 87 | | |
| | hexagonal | 10–42 | *Fusarium keratoplasticum* | [58] |
| | rod | 8–38 | *Aspergillus niger* | |
| | hexagonal | 15.45 | *Alternaria tenuissima* | [59] |
| | quasi-spherical | 16–78 | *Periconium* sp. | [60] |
| | star | 50–80 | cyanobacterium *Nostoc* sp. | [61] |
| AgNPs | spherical | 35–42.5 | cacao powder | [62] |
| | cubic and spherical | 30–90 | leaves of *Juniperus procera* | [63] |
| | spherical | 28 | tea polyphenol | [64] |
| | | 35 | Nyctanthes arbor-tristis | [65] |
| | circular | 100 | *C. cladosporiodes* | [66] |
| | triangular | 25 | *Talaromyces purpureogenus* | [67] |
| | spherical | 30–50 | coral | [68] |
| AuNPs | spherical | 14–17 | Macrotyloma uniflorum | [69] |
| | spherical | 3–45 | leaves of Bacopa monnieri | [70] |
| | cubic | 20–50 | flower of Butterfly tree | [71] |
| | triangular, hexagonal, pentagonal, rod, and truncated | 25–100 | *Ipomoea carnea* | [72] |
| | triangular | 600–800 | pumpkin | [73] |
| | spherical and narrow | 45–75 | *Abelmoschus esculentus* | [74] |
| CuNPs | spherical, hexagonal, cylindrical, triangular, and prismatic shapes | 10–50 | leaves of *Hagenia abyssinica* | [75] |
| | spherical | 50–250 | leaves of Magnolia kobus | [76] |
| | spherical | 2–10 | *Celastrus paniculatus* | [77] |
| PdNPs | spherical | 50–150 | *Prunus × yedoensis* | [78] |
| TiO$_2$ NPs | different | 200 | *Sesbania grandiflora* | [79] |
| | spherical | 15–20 | jackfruit leaves | [80] |
| | spherical | 20–90 | *Trigonella foenum-graecum* | [81] |
| MgO NPs | spherical and cylindrical | 67.70 | *Persimmon* | [82] |
| FeO NPs | honeycomb | 30–100 | flowers of *Avecinnia marina* | [83] |
| | spherical | 58 to 530 | *Amaranthus spinosus* leaf | [84] |

The most common metal nanoparticles produced via green synthesis are oxide and dioxide, including polymer-coated iron [85], zinc [86], copper [87], gold [88], silver, and their oxides [27,89,90]. Many plant species are used in the production of metal nanoparticles (Table 1). The fruits of langsat (*Lansium domesticum*) and amla (*Phyllanthus emblica*) are used for the production of gold nanoparticles with antimicrobial activity. In turn, AgNPs are rapidly synthesized with the use of *Calendula officinalis* and *Capsicum annuum* seeds. Tea plant (*Camellia sinensis*) leaves serve as a catalytic agent in the synthesis of ZnONPs [3]. Younis et al. [91] used dried *Rosa floribunda* petals for the green synthesis of MgONPs. Asiya et al. [92] used lemon fruit (*Citrus limon*) to produce gold nanoparticles. Another promising

trend in the production of nanoparticles is the use of biowaste, contributing to a reduction in its amounts. This approach has been reported by Jain et al. [93], who used the skin of jackfruit (*Artocarpus heterophyllus*) to isolate iron nanoparticles.

## 4. Antimicrobial Properties of Metal Nanoparticles

The direct toxic effects of metal nanoparticles on organisms depend primarily on the type and surface reactivity. Indirect effects are due to physical confinement, the release of toxic ions, or from the generation of reactive oxygen species (ROS). ROS are thought to lead to additional cellular responses classified as defense, proinflammatory effects (in plant and animal cells), and cytotoxicity [94,95]. Indirect toxicological effects of metal nanoparticles may include: oxidative stress and inflammation associated with ROS production, glutathione depletion and accumulation of oxidized glutathione in response to ROS production, protein denaturation, membrane damage, DNA damage, immune system reactivity and foreign body granuloma formation, and a reduction or loss of photosynthetic activity in algae and plants [95,96]. Much recent research has been directed toward developing self-responsive systems, e.g., triggered by specific physiological environments (local acidity, presence of substances secreted by bacteria) or external stimuli (photothermal, electrical, magnetic) [23,24,97,98].

### 4.1. Antibacterial Activity

The antibacterial activity of metal nanoparticles depends on their size, chemical composition, surface structure, solubility, shape, and tendency to aggregate, as well as environmental conditions, e.g., pH, salinity, and the presence of organic matter. The cell wall of Gram-positive bacteria is a complex mixture of glycopolymers and proteins. It consists of a peptidoglycan sacculus composed of long polysaccharide chains linked via peptide bridges. The wall surrounds the cytoplasmic membrane. It plays an important role in bacterial physiology, as it maintains the shape and integrity of the cell during growth and division. Additionally, it acts as an interface between the bacterium and its environment [99]. The bacterial cell wall is mostly composed of murein, which is multilayered in Gram-positive bacteria. In turn, Gram-negative bacteria have a single layer of murein additionally surrounded by an outer membrane. The potential mechanisms of the antibacterial effect of metal nanoparticles consist of their impact on the structure and function of cell membranes, disturbances in metabolic processes caused by the generation of reactive oxygen species, and a negative influence on the activity of enzymes and DNA synthesis [100]. As suggested by Wong et al. [101], nanoparticles should be smaller than 50 nm to become effective antimicrobial agents. Table 2 shows examples of the antimicrobial activity of metal nanoparticles.

The antiseptic capacity against eukaryotic cells is different. The feature that distinguishes eukaryotic cells from prokaryotic cells (bacteria, archaeons) is a very complex internal structure. Among other things, eukaryotic cells have a cytoskeleton. Nanoparticles, immobilized in a gel matrix, can exert antimicrobial activity by contacting the bacterial membrane, while they cannot be taken up and internalized by eukaryotic cells [102]. Gold and zinc nanoparticles in the form of colloidal solutions that allow penetration into cells may be effective against eukaryotic cells [103,104]. In the case of fungi, which belong to the group of eukaryotic organisms, the same interactions between nanometals and cells occur as in the case of bacteria [87].

Another important consideration is the long-term stability of nanoparticle surfaces and the ability to maintain antibacterial properties over time. A promising strategy is so-called smart antimicrobial surfaces, which can kill bacteria attached to their surface and then release dead bacteria and other contaminants on demand to reveal a clean surface under the right stimulus [105].

**Table 2.** Antimicrobial activity of metal nanoparticles.

| Metal | Pathogens | Nanoparticles | Genus or Species | References |
|---|---|---|---|---|
| **Ag** | Bacteria | AgNPs | *Acinetobacter baumannii, Coliform bacteria, Escherichia Coli, Klebsiella pneumoniae, Listeria monocytogenes, Salmonella enterica, Salmonella typhi, Staphylococcus aureus, Vibrio cholerae, Pseudomonas aeruginosa* | [62,106–111] |
| | Viruses | | Adenovirus type 3 (Ad3), Chikungunya virus (CHIKV), CoxB4, HAV-10, HSV-1, Hepatitis B virus (HBV), Herpes simplex virus (HSV), Human immunodeficiency virus (HIV), Influenza virus, Poliovirus, Respiratory syncytial virus (RSV), SARS-CoV-2 | [112–120] |
| | Fungi | | *Candida albicans, Candida parapsilosis, Candida tropicalis, Microsporum canis, Trichophyton mentagrophytes,* | [121,122] |
| | | AgNPs, AgClNPs | *Aspergillus flavus, Aspergillus niger, Candida albicans, Cladosporium cladosporioides* | [66,68,123] |
| **Au** | Bacteria | AuNPs | *Corynebacterium pseudotuberculosis, Escherichia coli, Streptococcus pneumoniae* | [124–126] |
| | | Gold-chitosan hybrid nanoparticles | *Pseudomonas aeruginosa, Staphylococcus aureus* | [127] |
| | Viruses | AuNps | Dengue, Herpes simplex virus (HSV), Human papillomavirus, Influenza, Lentivirus, Measles virus (MeV), Respiratory syncytial virus (RSV), Vesicular stomatitis virus (VSV) | [128–131] |
| | Fungi | | *Aspergillus flavus, Aspergillus niger, Candida albicans, Trichophyton rubrum* | [132] |
| **Bi** | Bacteria | BiNPs | *Enterococcus faecalis* | [133] |
| **Cd** | Bacteria | CdONPs | *Staphylococcus aureus* | [134] |
| **Cu** | Bacteria | CuNPs | *Bacillus subtilis, Escherichia coli, Listeria monocytogenes, Pseudomonas aeruginosa, Staphylococcus aureus,* | [75,107,135] |
| | | CuONps | *Bacillus subtilis, Escherichia coli, Psedomonas aeruginosa, Staphylococcus aureus, Klebsiella pneumoniae* | [136,137] |
| | Viruses | CuINPs | Feline calicivirus (FCV), H1N1 influenza | [138] |
| | | CuSNPs | Human norovirus | [139] |
| | | $Cu_2ONPs$ | Hepatitis C virus (HCV) | [140] |
| | | CuONPs | Herpes simplex virus type 1 (HSV-1) | [141,142] |
| **Fe** | Bacteria | FeONPs | *Staphylococcus aureus* | [143] |
| | | $Fe_2O_3$ modified by Ag | *Escherichia coli* | [144] |
| | Viruses | FeONPs | Dengue virus, Influenza A subtypes: H1N1, H5N1, and H7N9, Rotavirus | [145–147] |
| | Fungi | FeONPs | *Alternaria alternate, Aspergillus niger* | [20] |
| **Mg** | Bacteria | MgONps | *Acinetobacter baumannii, Enterobacter cloacae, Enterococcus faecalis, Escherichia coli, Bacillus megaterium, Bacillus subtilis, Staphylococcus aureus, Staphylococcus epidermidis* | [148–150] |
| **Mn** | Bacteria | MnNPs | *Escherichia coli, Staphylococcus aureus* | [151] |
| **Ni** | Bacteria | $NiFe_2O_4NPs$ | *Escherichia coli* NCIM 2345, *Salmonella typhimurium* NCIM 2501, *Staphylococcus aureus* NCIM 5021, *Streptococcus pyogenes* NCIM 5280 | [152] |
| | | NiONPs | *Bacillus subtilis, Escherichia coli, Protius mirabilis, Staphylococcus aureu, Streptococcus pneumoniae* | [153,154] |

**Table 2.** *Cont.*

| Metal | Pathogens | Nanoparticles | Genus or Species | References |
|---|---|---|---|---|
| **Pd** | Bacteria | PdNPs | *Bacillus subtilis, Escherichia coli, Psedomonas aeruginosa, Staphylococcus aureus* | [78,155] |
| **Pt** | Bacteria | PtNPs | *Bacillus subtilis, Escherichia coli, Klebsiella pneumoniae, Psedomonas aeruginosa, Staphylococcus aureus* | [151] |
| | Fungi | | *Aspergillus* sp., *Candida* sp. | |
| **Sn** | Bacteria | SnO$_2$NPs | *Escherichia coli, Staphylococcus aureus* | [156–158] |
| | Fungi | | *Candida albicans* | [159] |
| **Ti** | Bacteria | TiO$_2$NPs | *Bacillus subtilis, Enterococcus faecalis, Escherichia coli, Klebsiella pneumoniae, Proteus vulgaris, Pseudomonas aeruginosa, Salmonella typhimurium, Staphylococcus aureus, Streptococcus faecalis, Yersinia enterocolitica* | [80,81,160] |
| | Viruses | | Herpes simplex virus (HSV), Human norovirus, Influenza virus (A/PR8/H1N1) | [161–163] |
| | Fungi | | *Aspergillus fumigatus, Arthrographis cuboid, Aspergillus niger, Candida albicans* | [81,164,165] |
| **V** | Bacteria | Nalidixic acid-vanadium (V-NA) nanoparticles | *Bacillus cereus, Escherichia coli, Psedomonas aeruginosa, Staphylococcus aureus* | [166] |
| **Zn** | Bacteria | ZnONPs | *Bacillus cereus, Bacillus subtilis, Campylobacter jejuni, Escherichia coli, Klebsiella pneumonia, Listeria monocytogenes, Pseudomonas aeruginos, Pseudomonas vulgaris, Salmonella* spp., *Staphylococcus aureus* | [58,108,167] |
| | Viruses | | H1N1 influenza virus, Herpes simplex virus type-1 (HSV-1) | [168,169] |
| | Fungi | | *Alternaria alternata* (ITCC 6531), *Aspergillus niger* (ITCC 7122), *Candida albicans* | [60,170] |
| **Zr** | Bacteria | ZrO | *Bacillus subtilis* MTCC 1305, *Escherichia coli, Pseudomonas aeruginosa* MTCC 2453, *Staphylococcus aureus* MTCC 3160 | [171] |

### 4.2. Antifungal Activity

Metal nanoparticles can be used as potential inhibitors of fungal microorganisms (Table 2). To achieve the antifungal effect, nanoparticles are most often produced with the use of physical and chemical methods and less often through biosynthesis. It has been found that the mechanisms of action of nanometals against fungi are similar to those involved in their antibacterial properties, e.g., the induction of oxidative stress through ROS generation, damage to the cell wall and membrane through direct contact with nanoparticles, and cellular uptake of nanoparticles and subsequent harmful interactions with the cell interior [172]. In some cases, nanoparticles induce the deformation of fungal hyphae. Fungicidal activity is exhibited by the nano-oxides of some metals as well [173].

### 4.3. Antiviral Activity

Due to their unique properties, metal and metal-based nanoparticles have a number of medical applications in the treatment of diseases or as antiviral coatings of biological materials. Nanoscale metals, e.g., copper, silver, and gold, exhibit potent biocidal properties [174,175]. Their antiviral activity described in the available literature [128,176–179] consists of:

- Modification of the viral surface to prevent penetration into cells;
- Generation of virus-degrading reactive oxygen species (ROS);

- Interactions with structures present on the virus surface, proteins, and nucleic acids to destroy the integrity of the virus and inhibit its vital functions (genome replication, protein synthesis);
- Cleavage of disulfide bonds between cysteine amino acid residues to denature and inactivate viral glycoproteins.

The mechanisms through which certain nanoparticles act on different viruses are highly variable and depend both on the reactivity of the nanoparticles and on the structure of the virus [178]. Nanoparticles can be designed to contain conventional antiviral properties and traits that are unique to nanosystems, i.e., a small size, a high surface-to-volume ratio, and multi-functionality. This makes them an ideal component for the creation of anti-viral surfaces. Poorly water-soluble and unstable compounds can be modified with nanocarriers to ensure a better solubility and stability in physiological conditions [7].

## 5. Antimicrobial Properties and Applications of Some Metal NPs

### 5.1. Silver Nanoparticles (AgNPs)

Nanosilver is one of the most widely used biocidal nanometals. In fact, ionic silver has been eliminated from use by silver nanoparticles. This is mainly related to the susceptibility of silver ions to the formation of precipitation complexes inactivating the metal. Moreover, nanosilver is potentially safer, and its bactericidal properties can be additionally enhanced by, e.g., the addition of stabilizers [180]. A new technique to exploit the biocidal capacity of nanosilver is to produce biological cells with functionalized surfaces. Nanoparticles stabilized with cationic polymers adhere electrostatically to microbial cells, forming a uniform monolayer on the cell walls. "Cyborg" cells are then used to deliver the nanoparticles to the pathogenic microorganisms [181].

One of the bactericidal mechanisms of action is the binding of silver ions released from the surface of nanoparticles to the cell wall. It has been demonstrated that silver nanostructures ranging in size from several to a few dozen nanometers induce perforations in the bacterial wall, leading to bacterial death [182]. Silver nanoparticles can also cause the detachment of the wall from the cell membrane in Staphylococcus aureus and Escherichia coli, causing annihilation of these bacteria [183]. In comparison with Gram-negative bacteria, Gram-positive bacteria exhibit a greater resistance to nanosilver, as they contain greater amounts of negatively charged murein, which binds silver cations in the cell wall, thus limiting the penetration of the metal into the bacterial interior. Through damage to the cell wall, nanoparticles disrupt the electron transport chain, which results in disturbances in metabolism. This is related to the ability of nanosilver to absorb oxygen and catalyze the oxidation reaction. Oxygen absorbed on the surface of silver ions removes the hydrogen atom from cysteine thiol (–SH) groups. The removal of hydrogen yields a –S–S– bond between the amino acids. When this reaction takes place inside the electron transfer channel, the channel is closed and the bacterium loses the cellular respiration function and, consequently, dies [184]. Silver nanoparticles penetrating the cytoplasm are able to catalyze intracellular reactions and interact with cellular structures. Through reactions with functional groups of sulfur amino acids, they are responsible for damage to receptor and transport proteins, as well as enzymes, which disrupts the proper cell functioning. The bactericidal properties of silver nanoparticles are also associated with the generation of reactive oxygen species (ROS). These reactive chemical entities contain oxygen atoms with an unpaired electron. They exert a detrimental effect on microbial metabolism [185]. It was shown that 15 nm silver nanostructures were responsible for an increase in the concentration of ROS inside the cells of nitrifying bacteria. The high concentration of reactive oxygen species exhibited a strong correlation with the degree of microbial growth inhibition.

Raczkowska et al. [186] demonstrated the temperature dependence of biocidal intensity for POEGMA188 and P4VP coatings containing nanosilver. Siriwardana et al. [187] reported the ability of silver nanoparticles to bind and cleave disulfide bonds. This mechanism can potentially be used to combat the SARS-CoV-2 virus. In the case of enveloped viruses, AgNPs inhibit the process of viral nucleotide copying in infected cells, which,

in turn, prevents virus replication [188]. With its biocidal properties, Ag excellently disinfects and protects contact surfaces [113,189]. In addition to pure silver nanoparticles, functional antiviral agents containing nanoparticles of silver compounds can be successfully produced, e.g., to combat SARS-CoV-2 [190]. Silver nanoparticles can also be used to produce self-sterilizing composite polymers, custom packaging, and contact surface sterilizers [191,192]. Silver nanoparticles, with their well-explored properties, are part of commercial packaging (e.g., Duretham, DuPont, Sumitomo, Metal Ferro, NovaCentrix, US Research Nanomaterials, NANOMAS, Heraeus, and Kodak) extending the shelf life of food products [193] (Table 3).

**Table 3.** Application of silver and copper nanoparticles for production of self-sterilizing materials.

| NPs | Concentration | Material | Patogens | References |
|---|---|---|---|---|
| **AgNPs** | 0.27 ppm | Polyhydroxyalkanoate bioplastic film | Surrogates, murine norovirus (MNV), Feline calicivirus (FCV) | [192] |
| | 1% (complete inactivation) | Polylactide plastic | *Salmonella* spp., Feline calicivirus FCV | [191] |
| | $AgNO_3$ (0.5 M, 35 mL) and ethylene glycol, 25% inhibition of infection | Graphene oxide sheets | Feline coronavirus (FCoV) | [194] |
| | Less than 1 mg per 10 mL | PLA/silver-based nanoclay (organomodified MMT) | *Salmonella* spp. | [195] |
| | 20 mg AgNPs per 150 mL | Agar nanocomposite films | *Escherichia coli*, *Listeria monocytogenes* | [38] |
| | MSF-Ag at 110 μg per mL of *S. aureus* and 70 μg per mL of *E. coli* | Mesoporous silica flakes (MSF) | *Escherichia coli*, *Staphylococcus aureus* | [196] |
| | 25 mg $L^{-1}$ (silver content 1.5 mg $L^{-1}$) | Monodispersed silica spheres | *Escherichia coli* | [89] |
| | 0.005 M $AgNO_3$ solution | Temperature-responsive POEGMA188 and P4VP grafted polymer brush coatings attached to a glass surface | *Escherichia coli* ATCC 25922, *Staphylococcus aureus* ATCC 25923 | [186] |
| | 0.04–1.18 silver nitrate (mol $L^{-1}$) | Poly(N-isopropylacrylamide) (PNIPAAm) glass surfaces | *Escherichia coli* | [197] |
| **CuONPs** | 0.1% or 0.05% | Electrospun poly(3-hydroxybutyrate-co-hydroxyvalerate) PHBV polymers | Murine norovirus | [198] |
| **Cu₂ONPs** | 2% | Face mask | Influenza A viruses (H1N1) | [199] |
| **CuNPs** | 0.01–0.15 g $L^{-1}$ $Cu^{2+}$ | Cotton and polyester substrates | *Staphylococcus aureus* ATCC 25923 | [200] |

*5.2. Titanium Dioxide Nanoparticles (TiO₂NPs)*

$TiO_2$ is a well-known and relatively cheap metal oxide used in the photocatalysis process. It has a number of desirable properties, e.g., it is an excellent inhibitor of UV-activated viruses [201]. In industry, $TiO_2$ nanoparticles are used in filters, cosmetics, toothpaste, and soaps; they have strong bactericidal properties, remove odors, and, in combination with silver, serve as antibacterial agents. Titanium oxide nanoparticles generated through green synthesis exhibit very good antibacterial activity [81]. However, in 2016, the EFSA outlined the need for further research into the safety of $TiO_2$. Since that year, $TiO_2$ has no longer been regarded as a safe food additive. Its genotoxicity and accumulation in the organ-

ism after ingestion cannot be ruled out. Nevertheless, there is no ban on the use of $TiO_2$ nanoparticles in the food industry [202]. There is ample evidence in the literature showing the effects of $TiO_2$NPs on such viruses as human norovirus [161], human influenza virus (A/PR8/H1N1) [162], and herpes simplex virus [163]. However, $TiO_2$ is activated only by UVA photons ($\lambda < 387$ nm), which may limit its use in rooms with dominant visible-range lighting. Nakano et al. [162] confirmed that $TiO_2$-coated glass exhibited virucidal activity. In turn, the reduction in influenza virus titers below the limit of detection at typical indoor lighting intensity (~0.01 mW·cm$^{-2}$) required an almost 24 h exposure. Park et al. [161] showed that $TiO_2$ combined with fluorine exhibited better virucidal activity. The authors suggest that the surface fluorination of $TiO_2$ reduces UV reflection and generates additional free OH– and $O_2$, which can damage viruses. The $TiO_2$ photodegradation mechanism is effective in the inactivation of both non-enveloped viruses (noroviruses) and enveloped viruses, such as HSV-1. The SARS-CoV-2 virus is an enveloped virus, similar to HSV-1, with a similar lipid bilayer shell, and may be susceptible to the effects of $TiO_2$ [202]. As reported by Srivastava et al. [187], titanium oxide nanoparticles are effective in killing Newcastle disease viruses (NDV). Similar studies were conducted by Mazurkova et al. [203], who demonstrated the antiviral effect of these nanoparticles against the influenza virus, even when no UV radiation was used. As suggested in their study, $TiO_2$NPs can inactivate viruses through the penetration and disintegration of their envelope. It has been shown that $TiO_2$ nanoparticle coatings are a promising solution for the production of sterile surfaces and prevention of virus transmission.

*5.3. Magnesium Oxide Nanoparticles (MgONps)*

MgONPs have many properties, e.g., antimicrobial activity, chemical inertness, thermal stability, photostability, electrical insulation, non-toxicity, good biocompatibility and biodegradability, high surface area-to-volume ratio, low production cost, UV-blocking ability, and photocatalytic activity [204]. Hence, they are widely applied in such areas as food packaging, pharmacy, and medical sciences [46]. Magnesium oxide nanoparticles exert an extremely potent antibacterial effect, and their safety has been confirmed by the FDA (Food and Drug Administration) [205]. Mg is an essential element for living organisms; nevertheless, its nanoxide form makes magnesium highly active. Its antimicrobial activity is based on ROS production and lipid peroxidation [206]. MgO nanoparticles can also adhere to cell walls and break peptide bonds [207]. As reported by Huang et al. [208], nano-MgO used both directly and as an additive to interior wall paint has more potent bactericidal activity than $TiO_2$. Moreover, nano-MgO is active even in the absence of radiation.

*5.4. Zinc Oxide Nanoparticles (ZnONps)*

Due to their unique properties, ZnO nanoparticles are used in medicine. There are many methods for ZnONPs synthesis and each yields nanoparticles with different morphological structures determining their properties. The nanoparticles have been shown to have antibacterial activity against a wide range of microorganisms, e.g., *Escherichia coli*, *Pseudomonas aeruginosa*, *Klebsiella pneumoniae*, *Pseudomonas vulgaris*, and *Campylobacter jejuni* [209]. Additionally, they are used as drug transporters, in cancer therapy, and as strong antibacterial agents [210]. The US FDA has registered ZnO nanoparticles as GRAS [21]; hence, they are regarded as safe for the skin and can be used as dietary supplements [211]. The high toxicity of ZnO to pathogens and its minimal impact on human health facilitates the use of its nanoscale form for the production of food packaging and plastics with antimicrobial properties. Zinc oxide can be blended with petroleum-based biodegradable PHBV polymers to improve food stability and safety [192]. The bactericidal activity of ZnONPs consists of destructive diffusion through the cell membrane leading to bacterial cell lysis [212]. Moreover, ZnONPs have similar properties to TiO and can act as an effective, low-cost photocatalyst in the presence of water and sunlight or UV radiation [213]. The highest antibacterial activity against *E. coli* and *S. aureus* was achieved through the synthesis of ZnO nanospheres, with an average diameter of approx. 30 nm and largest

specific surface area of 25.70 mg [214]. In addition, in the case of viruses, ZnO nanoparticles induce the inhibition of protease and viral RNA and DNA polymerase [169,215]. As demonstrated by AbouAitah et al. [215], a nanoformulation with zinc nanoparticles is a potentially safe and low-cost hybrid agent that can be used as a new alternative strategy against the SARS-CoV-2 virus. This has been confirmed in the study conducted by El-Megharbel [216] showing virucidal properties of low concentrations of zinc nanoparticles ($IC_{50}$ 526 ng/mL) against the SARS-CoV-2 virus. Biosynthesized ZnONPs have been found to exhibit antimicrobial activity against a wide variety of microorganisms (Table 2). This activity is associated with ROS production, cell membrane damage, and interactions of ZnONPs with intracellular components [217,218]. Schwartz et al. [219] have developed an antimicrobial hydrogel composite coating by mixing poly(N-isopropylacrylamide) with zinc oxide nanoparticles. The coating was effective in the complete inactivation of *E. coli* bacteria at a low zinc oxide concentration of 1.33 mM. Other studies have also shown that the process of coating the surface of ZnO nanoparticles with polyethylene glycol may increase their antiviral efficacy and reduce their in vitro cytotoxicity [169].

*5.5. Gold Nanoparticles (AuNPs)*

Gold is known for its stability and ability to form complexes with biomolecules that support its antiviral and antibacterial properties. One of the properties of gold-based nanoparticles is the ability to block viral surface receptors, preventing viruses from the attachment and penetration of host cells. With their virucidal activity, AuNPs may be used to prevent infections with MEV and other enveloped viruses, such as SARS-CoV-2 [128]. Many researchers have observed that the size of gold nanoparticles is important for their antiviral activity. Gold nanoparticles that are equal to or greater than viruses can act as effective cross-linkers, binding multivalently and entrapping many virions. In contrast, smaller gold nanoparticles are less effective in covering the surface of individual viruses [130,131,220]. AuNPs have bacteriostatic activity against various bacterial strains through two of the most important mechanisms: the inhibition of ATPase activity, which blocks energy metabolism, and the inhibition of transcription via the prevention of tRNA ribosome interactions [221]. There are few literature reports on the use of AuNPs for the production of biocidal surface coatings or personal protective equipment. This is most probably related to the relatively high cost of gold nanoparticles compared with the cost of other metals.

*5.6. Iron Oxide Nanoparticles (FeONPs)*

Iron oxide nanoparticles have many biomedical applications, from nanotherapy to imaging contrast agents [222]. They are characterized by a high biocompatibility and magnetic properties, which are used to induce local hyperthermia, e.g., in cancer therapy [223]. With their properties, FeONPs have been approved by the American Food and Drug Administration (FDA) and other European Union agencies [224]. These nanoparticles have antibacterial properties against both Gram-positive and Gram-negative pathogenic bacterial strains [225,226] and against such viruses as influenza virus [145], rotavirus [147], and dengue virus [146]. The main mechanism of the antimicrobial activity of iron nanoparticles is the oxidative stress generated by ROS. Superoxide radicals ($O_2^-$), hydroxyl radicals (-OH), hydrogen peroxide ($H_2O_2$), and singlet oxygen ($1O_2$) can damage bacterial proteins and DNA [225]. Researchers are investigating the potential use of iron nanoparticles for the inactivation of the SARS-CoV-2 virus. Abo-Zeid and Williams [226] carried out detailed theoretical analyses of the possibility of binding $Fe_3O_4$ (magnetite) and $Fe_2O_3$ (hematite) nanoparticles with the binding domain of SARS-CoV-2 protein receptors and the E1 and E2 glycoproteins of hepatitis C virus (HCV). Coating nanoparticles with, e.g., polyvinylpyrrolidone (PVP), polyethylene glycol (PEG), or glycine polymers can enhance their antiviral activity, increase their stability, and reduce their toxicity [227,228]. The degradation of rotavirus virion capsids has been induced by the mechanism of electrostatic adsorption

to large-surface nanoparticles [147]. This phenomenon can be employed in the design of surface coatings [229].

*5.7. Copper-Based Nanoparticles*

In contrast to some other nanoparticles (gold, silver), copper is an essential trace element for the human organism and can be used safely in small doses [230]. Due to their properties, copper nanoparticles and compounds have been applied as part of biocidal preparations. CuNPs have been found to exert numerous toxic effects on bacteria, e.g., reactive oxygen generation, lipid peroxidation, protein oxidation, and DNA degradation in *E. coli* cells [231]. The antiviral properties of cuprous iodide nanoparticles, which are devoid of brown color in contrast to other copper compounds, have been investigated. CuI nanoparticles have been shown to dissociate into $Cu^+$ and generate highly reactive hydroxyl $(OH^-)$ radicals in phosphate-buffered saline [138]. It is well known that ROS cause oxidative damage to biological molecules by inducing nucleic acid mutations and random amino acid modifications [232]. CuI nanoparticles may exert such an effect against SARS-CoV-2 and other similar viruses. Both Cu(I) and Cu(II) oxides have been shown to be effective antiviral agents and to have antibacterial activity against both Gram-positive and Gram-negative bacteria [233]. $Cu_2O$NPs are effective in the fight against the highly mutagenic and drug-resistant hepatitis C virus (HCV). These nanoparticles block HCV attachment to host cells, thus inhibiting viral entry [141]. There are many studies on the use of nanoparticles for the production of packaging and personal protective equipment. For instance, biodegradable polymers and filtering masks containing CuNPs oxides have been investigated in this respect [198,199]. Some personal protective equipment has been approved by the FDA or the US Environmental Protection Agency (EPA) and is now available on the market [234,235].

*5.8. Tin(IV) Oxide Nanoparticles ($SnO_2$NPs)*

Tin(IV) oxide nanoparticles ($SnO_2$NPs) are inexpensive, non-toxic, and exhibit a high electron mobility, high photosensitivity, and good stability [236]. They exert potent anti-fungal effects on *C. albicans*, as confirmed in numerous studies [237]. Moreover, promising fungicidal effects were achieved with the use of various concentrations of tin-doped indium oxide nanoparticles [158]. Rehman et al. [159] investigated the antifungal activity of $SnO_2$ nanoparticles. They reported minimal fungicidal activity against *C. albicans* at a concentration >16 mg/mL. Maximum effects were achieved on *Escherichia coli* as well.

## 6. Hybrid Surfaces with More Than One Type of Nanoparticles

Given the different mechanisms of the biocidal activity of nanoparticles, a promising approach is to create hybrid surfaces with more than one type of particles to increase the self-sterilization efficiency. Hodek et al. [238] developed a hybrid polymethyl methacrylate (PMMA) surface coating containing silver, copper, and zinc cations to combat viruses present on surfaces. The HIV-1 titer was reduced by 99.5–100% after only a 20 min exposure to the coating. Other viruses were reduced by 97% (dengue virus), 100% (herpes simplex virus), and 77% (influenza virus). The researchers evidenced that the coating stably released ions at a sufficient rate to elicit virucidal activity but was safe for human cells. Nangmenyi et al. [144] developed a hybrid material composed of Ag-modified $Fe_2O_3$ nanoparticles embedded in fiberglass. Within the first minute of contact with the $Ag/Fe_2O_3$ fiberglass, a 99% inactivation of the *E. coli* cultures was already observed. The combined use of metal nanoparticles may reduce the risk of the development of pathogen resistance via many different inactivation mechanisms [144].

## 7. Biocompatibility of Metal Nanoparticles

Biocompatibility is the property of a material or substance functioning properly in a living organism. This means that, in addition to being present in living tissues, it must fulfill certain functions [239]. Biocompatibility depends on many factors: the targeted delivery

mechanism, coating, biodegradability, chemistry of surface phenomena, structure, stability of nanoparticle colloidal solutions, and their ability to integrate into a cell or tissue without causing adverse effects (Figure 2). A lack of biocompatibility for nanoparticles means a disruption of cellular and tissue metabolism leading to toxic effects in the body [240].

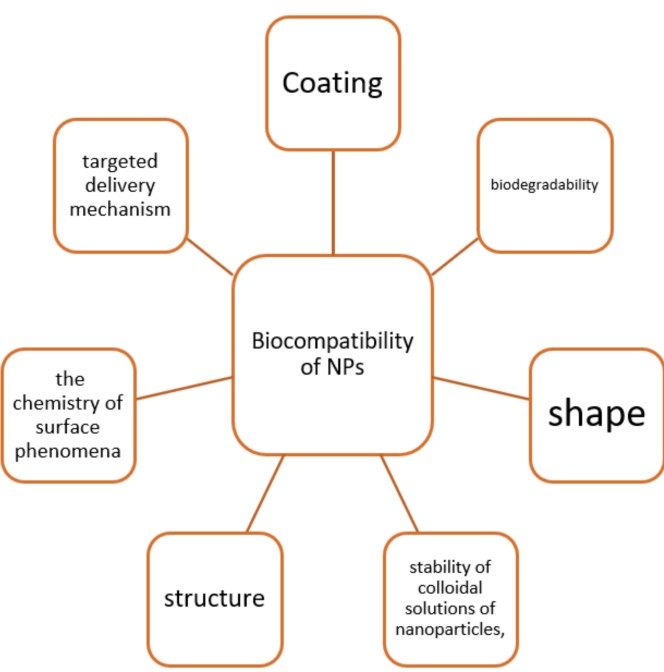

**Figure 2.** Chosen properties determining the biocompatibility of nanoparticles.

Nanoparticles show a higher biocompatibility than metal ions, which can attack both normal mammalian cells and pathogenic bacteria. The biocompatibility of antimicrobial agents with metal ions is often a major clinical problem. Pure metal nanoparticles with the highest biocompatibility include AuNPS. Compared with Ag and Cu nanoparticles, Au nanoparticles are much more biocompatible and therefore represent a potentially more therapeutically promising antimicrobial agent. In addition to pure metals, nanoparticles also exist in the form of metal oxides. Among the most biocompatible ones is ZnONPs. It is characterized by a low toxicity to living cells and can be successfully used for drug delivery, edible coatings, and, supportively, as a cosmetic additive [241]. Minimizing the toxicity of metal-based nanoparticles to normal eukaryotic cells without compromising their antimicrobial activity remains a major challenge. A future direction leading to an improved biocompatibility of nanoparticles may be the modification of their surface with polymers and peptides or polysaccharides [242,243].

## 8. Use of Metal Nanoparticles in Protective Edible Coatings

The appropriate packaging and coating of fresh fruit and vegetables is essential to prevent quick drying, loss of firmness, and exposure to pathogens. Coatings have an impact on water vapor permeability, which should be low in order to reduce the drying rate. Additionally, the oxygen permeability should be reduced to slow down respiration without the creation of anaerobic conditions promoting the emergence of bad taste and production of ethanol. Ideal coatings ensure resistance and durability during the distribution and transport of food products and high antimicrobial properties to minimize microbial growth [244]. Various active packaging systems are used to extend the shelf life of fruits and vegetables, e.g., oxygen and moisture absorbers, $CO_2$ emitters, and antioxidant/antimicrobial composites. In recent years, an increase in the use of bio/polymer nanocomposites based on nanoparticles (NPs) has been reported. The use of NPs in bio/polymers offers many advantages over other packaging systems. Submicron-

sized metal nanoparticles exhibit various possibilities for incorporation into food-grade polymer matrices serving as support systems. Metal nanosystems help to modify the properties of edible coatings. These properties are useful in solving food safety problems associated with pathogenic microorganisms, as they can limit the growth of yeasts, molds, and bacteria degrading the quality of food during storage and reducing the food shelf life [10]. Table 4 presents some studies on bio-nanocomposite films and edible coatings used to improve physicochemical properties of food and extend its shelf life.

**Table 4.** Examples of the use of metal nanoparticles in edible coatings.

| NPs | Coating | Food | References |
|-----|---------|------|------------|
| ZnONps | soy protein isolate | banana | [245] |
| | chitosan/gum arabic | | [246] |
| | chitosan | papaya | [247] |
| | carboxymethyl-cellulose | pomegranate fruit | [248] |
| | | tomato | [249] |
| | chitosan/gum arabic | avocado | [250] |
| | xanthan gum | tomato and apple | [251] |
| | chitosan/alginate | guava | [252] |
| AgNps | polylactic acid film | mango | [253] |
| | polyvinylpyrrolidone | asparagus | [254] |
| | carboxymethyl cellulose (CMC)-guar gum | strawberry | [255] |
| | sodium alginate-gum | black grapes | [256] |
| TiO$_2$NPs | chitosan/polyvinyl alcohol | cheese | [257] |
| | chitosan | tomato | [258] |
| Zn-MgONps | alginate solution | cold-smoked salmon | [259] |
| AgTiO$_2$NPs | chitosan | fruit | [260] |

The use of metal nanoparticles in edible coatings poses several problems that have to be solved. One of them is associated with the achievement of uniform dispersion of NPs and the problem of the agglomeration of nanoparticles [261]. Edible coatings must be appropriately applied to the raw material. The coating process consists of wetting the food surface with the coating solution followed by an adhesion step, in which, the coating solution penetrates the fruit or vegetable peel. Therefore, a high affinity between the coating solution and the food surface is necessary to facilitate the spread of the coating over the surface. Dipping, brushing, spraying, pan-coating, and fluidized the bed coating are well-known techniques, while electrostatic spraying has been used more recently [262].

Another issue is the undesirable effect of metal nanoparticles on the human organism [263]. Some nanoparticles have been approved for consumption, e.g., ZnONPs, which have been classified as GRAS by the FDA (21CFR73.1991) and as a safe additive (E6) by the EFSA [264]. Silver is on the EFSA list of the Scientific Committee for Food (SCF), i.e., a group of additives with no established daily intake recommendations but with an accepted level limited to 0.05 mg Ag/kg food, based on a maximum concentration that does not cause any detectable adverse changes in morphology, functional capacity, growth performance, or survival in humans [265].

Recently, due to the suspicion of some metal nanoparticles as carcinogens, many studies on non-toxicity and biocompatibility with the human body have been generated [266]. Tan et al. [267] investigated a polysaccharide coating based on metallic nanoparticles that was used as a packaging material to encapsulate curcumin. According to the authors, excellent biocompatibility properties were obtained. Pandey et al. [268] tested a protein-based

silver nanoparticle coating and found its low toxicity, effective application as a coating material for food products, and its high biocompatibility. Gnanasekar et al. [269] developed a material containing palladium nanoparticles that was safe for use in food and did not interact with red blood cells. It also destroyed bacterial cells and showed anti-cancer properties. Li et al. [270] used AuNPs to produce a β-glucan-based coating that was applied to edible mushrooms. This coating had beneficial effects in the gastrointestinal tract by causing the growth and activity of gut microbiota.

One of the trends in the latest research is the use of mixed nanomaterials, such as Zn-MgO, which is formed through association of zinc with highly biocompatible magnesium oxide (MgO) nanoparticles. The new material has been shown to retain strong antibacterial activity against Gram-positive bacteria, similar to ZnONPs, and to be safe for mammalian cells, similar to nano-MgO [259].

When analyzing the suitability of metal nanoparticles for commercial application in the production of self-sterilizing surfaces, several issues must be considered. Modified surfaces or new biocidal formulations in general face some serious challenges before they reach the market. Problems such as manufacturing costs, production scalability, intellectual property issues, or regulatory aspects are just some of the major hurdles that these products will have to overcome in order to be commercialized [271]. Some products, even with great lab results, fail when introduced on a larger scale due to high production costs, making them unattractive from a commercial standpoint. On the other hand, studies on the effects of a new product on human health are very time- and cost-intensive. However, the benefits of such materials may influence attempts to commercialize them [272].

## 9. Conclusions

1. Self-sterilizing coatings are widely used in many areas where pathogenic bacteria, viruses, and fungi are present. Their use seems to be one of the potential solutions to the problem of increasingly frequent global epidemics;
2. Zinc oxide nanoparticles are currently most widely used for surface coating due to their unique low cytotoxicity towards human cells, ROS-generating photocatalysis, and efficacy in damaging lipid membranes through diffusion and accumulation;
3. Similar to Ag and Cu, gold nanoparticles exhibit antiviral activity through binding to viral surface structures and thus inhibiting their function;
4. Iron oxide nanoparticles are characterized by a potent virus inhibition capability and biocompatibility, which can be increased via the modification of the polymer surface or hybridization with other particles. Moreover, they exhibit magnetic activity. Encapsulated iron and zinc nanoparticles have a greater antiviral activity and lower toxicity than non-encapsulated nanoparticles;
5. Given the evidenced effectiveness, low cytotoxicity, and relatively low cost of production, it is advisable to incorporate copper oxide NPs into the structures of self-sterilizing materials, including biodegradable polymers. $Cu_2O$-NPs are a promising option for the design of a new antiviral agent, and prompt further research into their practical application;
6. Some non-toxic nanometal particles are used for the production of edible coatings to prevent food spoilage.

**Author Contributions:** Conceptualization, D.G. and M.G.-K.; writing—original draft preparation, M.G.-K.; writing—review and editing, D.G.; visualization, M.G.-K.; supervision, D.G. All authors have read and agreed to the published version of the manuscript.

**Funding:** This research received no external funding.

**Institutional Review Board Statement:** Not applicable.

**Informed Consent Statement:** Not applicable.

**Data Availability Statement:** Not applicable.

**Conflicts of Interest:** The authors declare no conflict of interest.

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
