# Peer review of "Application of Metal Nanoparticles for Production of Self-Sterilizing Coatings"

_coatings, doi:10.3390/coatings12040480_

Round 1
Reviewer 1 Report
In the Review "Application of metal nanoparticles for production of self-sterilizing coatings" interesting and actual facts about self-sterilizing coatings based on metal nanoparticles are presented. Review can be published in Coatings mdpi after major revision. The following points should be mentioned.
1. It will be useful to add a Table including information on methods of the synthesis, shape, and size of the nanoparticleÑ–.
2. Mechanisms of the biological activity of the nanoparticles are sparingly described. Appropriate discussion about the direct and indirect impact of nanoparticles on microorganisms should be provided.
3. Information on the stimuli-responsive antibacterial coatings with embedded nanoparticles which able to self-cleaning is absent in the Review. It is an absolutely new trend which is very popular in recent times.
4. Appropriate discussion on the toxicity of the nanoparticles on eukaryotic cells should be provided.
5. Chemical methods are not suggested in details in Figure 1.
6. I suggest to cite relevant publications where similar results were presented.
https://doi.org/10.1016/j.colsurfa.2022.128525
https://doi.org/10.1016/j.msec.2019.109806
https://doi.org/10.1021/acsami.7b13565
Author Response
The authors thank the reviewer for evaluating the paper and providing important comments. By making these corrections to the manuscript, there will be an increase in the scientific value of the article.
- It will be useful to add a Table including information on methods of the synthesis, shape, and size of the nanoparticleÑ–.
Due to the huge number of works it is not possible to add all the required information. According to scopus database only for the query “gold nanoparticles” there are 97254 papers. A table with examples of particles obtained by the method of green synthesis was added in the paper
- Mechanisms of the biological activity of the nanoparticles are sparingly described. Appropriate discussion about the direct and indirect impact of nanoparticles on microorganisms should be provided.
The manuscript has revised.
- Information on the stimuli-responsive antibacterial coatings with embedded nanoparticles which able to self-cleaning is absent in the Review. It is an absolutely new trend which is very popular in recent times.
The manuscript has revised.
- Appropriate discussion on the toxicity of the nanoparticles on eukaryotic cells should be provided..
The text has been supplemented in many places with the required information. In part, the text already contained a lot of information about the toxicity of nanoparticles to fungi. Fungi are organisms composed of eukaryotic cells.
- Chemical methods are not suggested in details in Figure 1.
The figure has been corrected
- I suggest to cite relevant publications where similar results were presented.
https://doi.org/10.1016/j.colsurfa.2022.128525
https://doi.org/10.1016/j.msec.2019.109806
https://doi.org/10.1021/acsami.7b13565
The articles have been cited.

Reviewer 2 Report
In the manuscript entitled “Application of metal nanoparticles for production of self-sterilizing coatings” authors are reporting on preparation methods of the metal nanoparticle, their antibacterial, antifungal and antiviral activity focusing in the end on their application in self-sterilizing coatings.
At the moment there are a large variety of studies related to the topic of nanomaterials with antimicrobial applications which makes it harder for readers to be able to access all of them. Making a review focused on a specific application can help the readers in the future. However some observation can be made.
The authors have addressed very well the topics regarding the nanoparticles aims as well as their preparation methods. They have then presented briefly some examples of NPs with various activity: antibacterial, antifungal and antiviral.
An important characteristic of the nanoparticles especially for those that are used as protective edible coatings, is their biocompatibility.
Although the authors have mentioned for the nanoparticles that are known to be biocompatible, this is true with some limitations. Considering this, at least a subchapter where the biocompatibility is a bit more discussed should be added. Also, for the section 7, it should be added if the studies also report the biocompatibility of the coating.
Finally, there are some duplication of some of the references. For example 131 is similar with 133, 144 with 146 and 151. The authors need to check the references and make the appropriate changes in the text as well.
Author Response
The authors thank the reviewer for his insightful and useful comments.
Although the authors have mentioned for the nanoparticles that are known to be biocompatible, this is true with some limitations. Considering this, at least a subchapter where the biocompatibility is a bit more discussed should be added. Also, for the section 7, it should be added if the studies also report the biocompatibility of the coating.
These comments were taken into account in the manuscript
Finally, there are some duplication of some of the references. For example 131 is similar with 133, 144 with 146 and 151. The authors need to check the references and make the appropriate changes in the text as well.
The literature list has been revised.

Reviewer 3 Report
Reviewer comments
This manuscript describes “Application of metal nanoparticles for production of self-sterilizing coatings”. This is an interesting and well written review article on various applications of metal nanoparticles in self-sterilizing coatings. However, this is some issue in the current manuscript, and can be consider for publication after addressing following concerns.
Major and Minor concerns:
- In table 1, I think authors are missing many important examples in this list. This table need more attention and need to have other important articles in this filed.
- In table 2, authors are also missing many important examples in this list too. Need to be more comprehensive.
- Ideal properties and criteria of commercially viable self-sterilizing coatings of metal nanoparticles should also be summarized in this review.
- All references should be uniform.
- Some more recent references can be added in introduction part.
Author Response
Thank you for reviewing our manuscript. Your comments were very insightful and allowed us to significantly improve the quality of our manuscript. We hope that the corrections to the manuscript and the accompanying responses are sufficient to make our manuscript suitable for publication in Coatings.
In table 1, I think authors are missing many important examples in this list. This table need more attention and need to have other important articles in this filed.
In table 2, authors are also missing many important examples in this list too. Need to be more comprehensive.
The Scopus database for the query “metal nanoparticles” throws up 169,632 articles. Placing even the most important examples in Tables 1 and 2 exceeds the volume of a single article. Hence, only examples of antimicrobial activity of metal nanoparticles and application of nanoparticles for production of self-sterilizing materials are included in the tables. However, as suggested by the reviewer, the tables have been partially completed.
Ideal properties and criteria of commercially viable self-sterilizing coatings of metal nanoparticles should also be summarized in this review.
The manuscript has supplemented with the above information.
All references should be uniform.
References have been corrected
Some more recent references can be added in introduction part.
The following literature items have been added:
- Jung, S.; Myung, Y.; Das, G.S.; Bhatnagar, A.; Park, J.W.; Tripathi, K.M.; Kim, T. Carbon nano-onions from waste oil for application in energy storage devices. New J. Chem. 2020, 44, 7369–7375, doi:10.1039/d0nj00699h.
- Lyagin, I.; Stepanov, N.; Frolov, G.; Efremenko, E. Combined Modification of Fiber Materials by Enzymes and Metal Nanoparticles for Chemical and Biological Protection. Int. J. Mol. Sci. 2022, 23, 1359, doi:10.3390/ijms23031359.
- Chandrakala, V.; Aruna, V.; Angajala, G. Review on metal nanoparticles as nanocarriers: current challenges and perspectives in drug delivery systems. Emergent Mater. 2022, 1, 1–23.
- Moradi, M.; Razavi, R.; Omer, A.K.; Farhangfar, A.; McClements, D.J. Interactions between nanoparticle-based food additives and other food ingredients: A review of current knowledge. Trends Food Sci. Technol. 2022, 120, 75–87.
- Hansjosten, I.; Takamiya, M.; Rapp, J.; Reiner, L.; Fritsch-Decker, S.; Mattern, D.; Andraschko, S.; Anders, C.; Pace, G.; Dickmeis, T.; et al. Surface functionalisation-dependent adverse effects of metal nanoparticles and nanoplastics in zebrafish embryos. Environ. Sci. Nano 2022, 9, 375–392, doi:10.1039/d1en00299f.
- Parveen, K.; Banse, V.; Ledwani, L. Green synthesis of nanoparticles: Their advantages and disadvantages. In Proceedings of the AIP Conference Proceedings; AIP Publishing LLCAIP Publishing, 2016; Vol. 1724, p. 020048.
- Shevtsova, T.; Cavallaro, G.; Lazzara, G.; Milioto, S.; Donchak, V.; Harhay, K.; Korolko, S.; Budkowski, A.; Stetsyshyn, Y. Temperature-responsive hybrid nanomaterials based on modified halloysite nanotubes uploaded with silver nanoparticles. Colloids Surfaces A Physicochem. Eng. Asp. 2022, 641, 128525, doi:10.1016/j.colsurfa.2022.128525.
- Hadzhieva, Z.; Boccaccini, A.R. Recent developments in electrophoretic deposition (EPD) of antibacterial coatings for biomedical applications - A review. Curr. Opin. Biomed. Eng. 2022, 21, 100367.

Round 2
Reviewer 1 Report
The quality of the manuscript was essentially improved after revision. The manuscript can be accepted for publication in its present form.
Reviewer 2 Report
The authors have made the changes asked and the quality of the manuscript improved. The paper can be accepted in the present format.
Reviewer 3 Report
Reviewer comments
This manuscript “Application of metal nanoparticles for production of self-sterilizing coatings”. Authors have partially addressed reviewer concern however overall quality of manuscript has been improved. Therefore, this work can be consider for publication .